# The Social Utility and Health Benefits for Older Adults of Amenity Buildings in China’s Urban Parks: A Nanjing Case Study

**DOI:** 10.3390/ijerph17207497

**Published:** 2020-10-15

**Authors:** Binyu Liu, Ye Chen, Meng Xiao

**Affiliations:** 1Digital Landscape Architecture Lab of Southeast University, Landscape Architecture Department, School of Architecture, Southeast University, Nanjing 210096, China; 220180070@seu.edu.cn; 2China Construction Science & Technology Group Co., LTD., Shenzhen 518000, China; xiao.meng@cscec.com

**Keywords:** amenity building, social interaction, elderly-oriented, landscape, behavioral pattern

## Abstract

As China’s population rapidly ages, research and discussion on how to better optimize public spaces for the elderly’s health and benefit continue to deepen. This study uses observational surveys and questionnaires to investigate the elderly visitors of Nanjing’s urban parks and explore the impact the parks’ amenity buildings (structures built to provide visitors with conveniences, e.g., shelters and pavilions) has on their health and associated socialization tendencies. Data were collected from ten amenity buildings in ten separate parks to compose a total dataset of 728 activity statistics and 270 valid questionnaires. The study’s results indicate that amenity buildings significantly increase opportunities for older adults to socialize and thereby can increase this demographic’s associated health benefits. The social activities formed around amenity buildings are found to improve social interactions and connectedness among older adults more compared to other age groups. Elderly participation in social activities is also found to positively correlate with environmental characteristics. High-quality landscapes ensure healthy development of social activities within amenity buildings and promote the occurrence and continuation of social interactions. In order of highest to lowest impact on elderly activities, the following factors were identified and scored: amenity building scale, lighting, comprehensive surrounding environment, surrounding amenities, water features, and vegetation. This research also reveals that among existing amenity buildings, there is insufficient support for certain activities and therefore, parks need to be improved to address this deficiency. Overall, this study indicates that under China’s current aging trends, amenity buildings have become an especially important infrastructure within urban public space, and their design trend is to incorporate the dual characteristics of “recreation + society”.

## 1. Introduction

In recent years, China’s population has entered a period of rapid aging. According to data from the National Bureau of Statistics, by 2018, the proportion of the country’s population that was aged 60 years and older had reached 17.9%, far exceeding the 7% standard set by the United Nations [1]. It is estimated that by 2050, China’s population who will be aged 60 and older will reach 479 million people, accounting for 35.1% of the total population [2]. Additionally, the 2015 China General Social Survey (CGSS) [3] indicated that more than 70% of the national population is in a transitional state between health and disease, namely a state referred to as “suboptimal health status” (SHS). Among this large demographic, older adults are a high-risk group of SHS, which can be further divided into the three SHS areas of physiology, psychology, and social adaptability. In all three areas, CGSS reports show that the elderly population is facing major challenges, with only 13% of this group reporting good physical health, 20% reporting being in desperate psychological conditions and always feeling depressed, and 23% considering themselves unhappy or very unhappy.

Studies have found a significant positive correlation between the elderly population’s objective health status and their subjective wellbeing. As such, it seems improving this demographic’s quality of life is the key to enhancing their subjective wellbeing. Towards this end, the value of urban parks and green spaces is now well-established. The existing literature shows that green public spaces provide visitors with a variety of physical and mental health benefits [4,5,6], and can help improve overall wellbeing [7,8,9]. Including spaces which encourage physical exercise, contact with natural scenery, and social interaction, actively adjusting these health-supporting environments to enhance the happiness of their elderly visitors [10] and create a sense of “place attachment” has become an important global development trend [11].

Across many disciplines, including public health, social sciences, and landscape architecture, researchers are studying how physical activity in public parks can better promote health [12,13,14,15], especially the health of older adults [16,17]. Additionally, research attention has gradually increased towards issues such as the elderly population’s social adaptability and how being socially active can improve health in comprehensive and robust ways. Meanwhile, other research has also shown that satisfying psychological needs may be more important for older adults than other age groups, and that this demographic tends to go to public parks more often [14]. Finally, urban parks and open green spaces have been shown to be able to provide positive social interaction opportunities and have a positive impact on longevity [18,19,20]. Taken together, this research clearly indicates that by actively adapting to society and participating in social activities, older adults can promote their health and thereby achieve a higher sense of wellbeing.

Due to changes in communities and health, the social networks of older people tend to become smaller over time [21,22], resulting in this demographic’s increased risk of social isolation [23]. Two forms of social isolation have been identified: first, there is social isolation characterized by a lack of social relations and low levels of social activity, and, second, there is social isolation characterized by perceptions of isolation, loneliness, and lack of social support. Many sociologists and psychologists associate the perception of having social support with physical and mental health [24,25,26]. Relatedly, a large-scale study has shown that older people with a greater number of social contacts report fewer depressive symptoms [27]. Additionally, research shows having social support systems can help dampen the impact of stressful life events which tend to lead people towards depression and other mental illnesses [28]. Compared with younger age groups, the relationship between social participation and health is stronger among the elderly population [29]. It has been proposed that this is because social engagement can provide older people with meaningful social roles, which thereby confers them a sense of value, purpose, identity, and attachment to their communities [30].

Other studies suggest that parks and open spaces have become the “third place” outside of work or home where older adults can engage in social activities [31,32]. Perceiving open spaces and parks as social venues affects older adults’ experience of pleasure and “sociality” more [14,33,34]. That is, it is thought such public spaces may be critical for establishing social identity and interactions between residents [35,36]. It is also noted that the existence and quality of public spaces are among the basic elements which affect an older person’s decision whether or not to remain in their current home or community long-term, also known as “aging in place” [37].

Older people are motivated to seek out social connections by visiting parks and green spaces [7], and parks must provide sufficient set recreational facilities in order to enhance these possible social interactions [36,38]. One US-based study [39] classified the facilities or amenities as physical activity facilities (e.g., tennis courts, fields), built amenities (e.g., barbecues, bathrooms, parking lots), and natural amenities (e.g., coastlines, green belts), as a classification result of referring to in-person park audit tools including the Community Park Audit Tool [40], the Environmental Assessment for Public Recreation Space tool [41], and the Public Open Space Tool [42]. Previous research [43] found built structures to be the landscape element with the least restorative effects on visitors, including water features, topography, vegetation, roads, and pavements. However, it has also been shown that in Hong Kong [44], amenity buildings (structures built inside parks to provide park visitors with shade, seating, and other conveniences, such as shelters and pavilions) have been proven to be popular with older adults. Organized groups often perform Eastern physical activities and entertainment here (e.g., Taiqi exercises, opera singing, etc.), while the amenity building where such activities are hosted has not been found to be common in Europe. Europeans are considered more likely to participate in individual activities, and make decisions based on personal considerations like self-efficacy, enjoyment, and attitude [45].

Through many decades of public park development across the world, it can be seen that Eastern and Western countries have distinct design tendencies and features, including those of their amenity buildings. In Western countries, more attention is often given to the functionality of these amenities. In contrast, these same buildings are usually called “landscape buildings” in China as their effect on the park’s landscape imagery is more valued. Integrating the location of the building with the natural environment aesthetically is also often considered of primary importance, as they are able to provide a scenic place for activities.

Placemaking has become a popular concept and the characteristics of a place and its quality of design can affect the behavior of people who live in it [46]. As such, amenities which provide adequate space and safety are more likely to promote social interaction. Researchers have shown that older people prefer stable places with plants, lighting, seats, and other structures to provide shelter from wind and sunlight [47,48]. Amenity buildings, which can provide a stable and safe space for activities, will be able to promote the occurrence, participation, and continuity of social activities among the elderly population, and are therefore of great significance towards maintaining their physical and mental health and community connections.

Visitor usage of Chinese parks is different from that of Western countries. In China, the elderly population is the largest user group of urban parks. Due to having a large amount of leisure time, the elderly population aged 50 to 80 has become the main user group of the landscape environment [18,49]. Older adults in Chinese cities have a high demand for urban parks and green spaces because of the high population density and lack of green spaces in traditional urban areas. There are often many spontaneous activity groups in urban parks at all levels in China [50]. Amenity buildings in China are not only designed for people’s activities but, in fact, offer several distinct types, such as verandas, pavilions, and corridors. Such sites used to be places of entertainment for China’s ancient literati, but now they often function as socially embedded spaces for the contemporary elderly population, while also carrying symbolic significance as a part of China’s cultural landscape heritage [51].

The purpose of this study is to help narrow the discourse gap between the previous research described above and current urban park development by evaluating the behavioral patterns and preferences of older adults in China’s amenity buildings via a case site. This study explores the influencing factors of such amenities and their behavioral affordances, and based on the survey results, provides a theoretical basis for using these spaces towards the advancement of the elderly population. In particular, we aimed to investigate the following questions:(1)What is the significance and function of amenity buildings for older adults?(2)What kind of amenity buildings complement the neighborhood space of the community?(3)What health-benefiting activities do older people engage in when visiting amenity buildings?(4)How do social connections work?(5)What are the implications of the above findings for the planning and design of urban parks and their amenities?

## 2. Methods

This study focuses on the behavior of older adults who engage in social activities in the amenity buildings of Nanjing’s public parks and explores the positive physical and psychological effects these activities may have on older adults, as well as the performance of such typical amenities in parks in terms of being age-friendly (Figure 1).

It should be noted that the scope of elderly participants in this study included those who were 50 years of age or older. We believe this is valid because people between 50 and 60 years of age are also among the population who has entered a period of later maturity [52]. China’s national conditions may be also be considered, with many Chinese women retiring at the age of 50 and subsequently, becoming the main users of many urban parks. It may also be worth noting that one study by Godbey and Blazey compared younger elderly groups (50–64 years old) to older elderly groups (65+ years old) and they did not find differences between the groups in terms of their satisfaction with park use for leisure [53]. Finally, we note it is common to use 50 years as the lower limit in studies on aging [54,55,56].

### 2.1. Selection of Parks and Amenity Buildings

Nanjing is located in the Yangtze River Delta of eastern China and has a humid, subtropical climate. The Qinhuai River is the largest local river in Nanjing and is known as “Nanjing’s mother river”. Xuanwu Lake is known as China’s largest “royal garden lake” and has long been an important place for the daily entertainment and recreation of the city’s citizens. Xuanwu Lake and Qinhuai River are both located along the border of Nanjing’s ancient city wall and traditional urban districts. Due to various historical developments, there are a large number of old residential areas nearby both these waterways and they are populated by many middle-aged and elderly residents. During our study’s initial observations of the parks along the Qinhuai River, we saw older adults were the main groups carrying out activities within the parks, and that they were engaging in a large variety of activities. In order to minimize any potential misrepresentation by any individual park’s survey, this study selected eight typical urban parks along the Qinhuai River which had large surrounding residential areas. Specifically, the eight parks included Xiuqiu Park, Gulin Park, Qingliang Mountain Park, Xiaotao Park, Wulongtan Park, Mochou Lake Park, Bailuzhou Park, and Yueya Lake Park. In addition, as further comparator sites, Xuanwu Lake Park and Heping Park were also included in the survey. They are, respectively, Nanjing’s largest and smallest city parks, and are both located in the downtown area (Figure 2).

To mitigate the impact of the parks’ varying levels of accessibility, a single amenity building was selected from each park to serve as our final survey site. The amenity building site selections were based on our initial observations of older people’s activities and locations within the park, as well as in consideration of the sites being located within an assumed comfortable walking distance of 500 m from park entrances. The final result was a total of 10 typical amenity buildings being selected, with each being free-to-enter, having open interior spaces, and being located within 500 m of their respective park entrances (Figure 3). Selected buildings included mainly pavilions, corridors, and related architectural combinations. Each selected amenity building also had a special type-name designated by the park, similar to “shelter”. As an example, please see the images of Wulongtan Park’s amenity building, dubbed the “Ouxiangxie Shelter”, below (Figure 4). This survey site was located in the middle section of its park’s long and narrow waterfront area, and was 248 m away from the entrance.

### 2.2. Selection of Survey Period

In Nanjing, the month with the highest Comfort Index, which is based on temperature and relative humidity, is May. During this late spring and early summer period, the city’s elderly population can be seen feeling comfortable and willing to go out more [57]. Subsequently, the survey period for our study was set as April to June 2018.

#### 2.2.1. Observation Times

Surveys were conducted of elderly groups engaged in social activities at each of the selected amenity buildings via on-site records and interviews. Trained students were selected, for varying weekdays and weekends, to complete the survey data collection, and used standardized forms to record the time, place, weather, people, activities, and other information of each observational survey. Overall, there were 20 groups of two students each and each group visited the 10 parks across 6 daily time periods to observe the sites at 06:00–08:00, 08:00–10:00, 10:00–12:00, 12:00–14:00, 14:00–16:00, and 16:00–18:00. For certain activities which had primarily activity hours, observation times were adjusted accordingly.

#### 2.2.2. Questionnaire

In addition to observational data, a questionnaire survey was completed by individual members of the elderly groups engaged in social activities. This was done at the same time the sites were observed. Individuals from the various groups were approached and the purpose of the questionnaire was then explained to them. All individuals who agreed to participate were invited to fill in the questionnaire. They could then choose to complete the questionnaire immediately via paper or they could complete it electronically by scanning a QR code with their mobile phone and completing the questionnaire at a later time. For any individuals who expressed difficulty in reading the questionnaire, the questions were asked orally by our student teams and their answers were recorded on the individual’s behalf.

The questionnaire included four sections (Table A1): demographic characteristics, activity preferences, cognitive description, and activity effects. The first section requests background information regarding the questionnaire taker such as their gender, age, residence, occupation, and other demographic characteristics. The second section focuses on older adults’ activity preferences related to their corresponding amenity building, including activity times, what groups they are engaged in, visit frequency, and social channel of activities, among other details. In the third section, a five-point Likert scale was used to evaluate older adults’ cognition of the amenity building’s environment. Finally, in the fourth section, a five-point Likert scale was used for their self-perceived health status after completing activities, as a measure of health-related quality of life (HRQoL) [58].

We noted several difficulties during the process of administering our questionnaires to the elderly participants, including that their memory, reading, and comprehension abilities were often limited. Their reading stamina was also low, and when they encountered questions that were not easily and quickly understood, they would often skip the question outright. These experiences led the research team to carefully adjust the design of the questionnaire and simplify many of the questions. The result was that the final questionnaire was limited to 20 questions which mainly focused on activities in the amenity buildings and the individual’s perceptions and feelings towards those activities.

### 2.3. Data Analysis

The activity data and questionnaire data were classified and organized with Excel, and SPSS^®^ (v22) software was used to evaluate and analyze the behavioral patterns and preferences of the elderly participants, as well as the potential health benefits of their activities. The influencing factors of each amenity building were scored in order to evaluate the potential relationship between different amenity buildings’ corresponding factors and health benefits for older adults. Subsequently, the Pearson correlation between these factors and the physical activity score was calculated.

The factors composing the environmental index were all regarding the physical features of the building, which were calculated and assessed by two professionals. Scores were based on a 10-point scale, where final scores would be between 1 and 10 points, to standardize the calculation in SPSS software. Finally, Pearson’s product moment correlation and Chi-square statistics were used to analyze the physical evaluation of the amenity building through objective indicators of the observational surveys and the subjective evaluation of the questionnaires. Through this statistical analysis, the influence of various factors on the elderly group activities in the amenity building, as well as the relationship between different types of activities and these factors, were reviewed.

### 2.4. Factors

The measured factors of amenity buildings consist of physical factors and activity factors. Physical factors include surrounding environment and amenity conditions, activity factors include types of activity, number of activity participants, and physical activity score (Table 1).

#### 2.4.1. Surrounding Environment Factors

Several previous studies have proposed that the value of nature is still most commonly examined through the visual signifiers of “green” and “blue” space such as aquatic environments, parks, and gardens [60,61,62,63,64,65,66]. Other studies have shown that water and plants are two important landscape elements of public spaces because they can help provide a comfortable environment by improving the space’s microclimate [67]. The significant contributions these two elements appear to make towards restorative experiences have also been widely reported in numerous studies [68,69,70].

Numerous studies have confirmed that water features provide the most restorative landscape [68,71], and aquatic areas are frequently included as aspects of people’s favorite places [72]. There are also studies on other specific landscape elements; for example, a study of health anxiety among older adults in Bulgaria found significant inverse associations with “awareness of nature experiences”, which included, among other factors, appreciation of bird songs and vegetation [73].

Informed by the previous literature and investigations, the quality of the amenity buildings’ surrounding environmental factors (Table 1) was evaluated based on the three dimensions of water, vegetation, and comprehensive surrounding environment (CSE). The water factor was scored based on the percentage of surface area shared between the water and the amenity building. The vegetation factor was scored based on the amenity building’s density of vegetation or proportion of vegetation coverage. Finally, a CSE factor was scored using the site’s average score of water and vegetation factors, and added additional points if the site included special landscape elements, including environmental sounds, e.g., birdsongs, or special water features, such as fountains. Including the CSE factor towards the overall environmental index score was done to strengthen the weight given to the overall sense of the site and attempt to evaluate the site’s environmental quality from a more comprehensive perspective.

#### 2.4.2. Amenity Buildings Factors

Previous research has demonstrated the importance a park’s amenities can make towards a local parks’ attractiveness to elderly visitors. Correspondingly, a lack of appropriate amenities, such as toilets and cafes, or other simple amenities such as benches, can represent significant impediments to walking and other uses of outdoor spaces [74]. Some studies have confirmed that regardless of park size, the presence of general amenities (such as BBQs, drinking fountains, lighting, public toilets, rubbish bins, directional signage) can be significantly and strongly associated with high park use [42,75,76,77].

Based on the previous literature and related field investigations, this study selected three physical factors of its amenity buildings to evaluate their general quality and function: building scale, surrounding amenities, and lighting (Table 1).

The scale factor was scored according to the square meter area of the amenity building. The score took into account people’s typical social distance of 1.2 m, which implies one person uses approximately 2~3 m^2^ with a radius of 1.2 m. This means a 90 m^2^ amenity building can accommodate approximately 30 people. Generally, in this study, the area of parks’ amenity buildings was not more than 100 m^2^, so if the building area exceeded 90 m^2^, it would be calculated with a full score for this factor.

The score for the surrounding amenities factor was determined by the type and number of certain amenities within or nearby the amenity building. These included having seats, tables, rubbish bins, drinking fountains, vending machines, toilets, storage boxes, and bulletin boards. It is noteworthy that for the elderly population, having a bathroom at or nearby the amenity building is quite indispensable. We assumed the average walking speed of an elderly visitor is 3.2–3.9 km/h, and that their maximum endurance time to walk to a bathroom from the amenity building is 5 min. Therefore, amenity buildings with a bathroom within 250 m were considered as one dimension of this factor’s score. Depending on how they may support social interactions, seats were rated on three levels. One point was granted for having each of the other amenities. For example, an amenity building with four seats (3), a toilet nearby (1), and a rubbish bin (1) would receive a surrounding amenities factor score of 5.

Lighting scores are based on the total scores of five normal types of lighting, including lighting inside corridors, garden lighting, lawn lighting, building contour lighting, and nearby street lighting.

#### 2.4.3. Physical Activity Score

Activity intensity is one sign of activity healthiness and the amount of energy required for different types of physical activity types varies. Currently, widely recognized indicators of physical activity intensity are the Metabolic Equivalents of Energy (METs), which refers to the degree that one’s metabolic rate rises during exercise as compared to one’s metabolic rate at rest. Based on the various health indicators of older adults involved in sports of different intensities proposed by Siscovick [78], this paper primarily refers to the metabolic equivalent table provided in “Sports Nutrition”, edited by Ronald [79], to classify the assumed activity intensity of older adults during their observed activities (Table 2): (1) Low-intensity activities are related to behaviors such as sitting, standing, and observing; (2) Medium-intensity activities are related to singing, walking, and mild physical exercise; (3) High-intensity activities are related to running, jogging, or any other vigorous physical exercise. Due to the limitations of space and physical conditions of older adults, the only high-intensity activity assumed in the amenity building is dancing.

A physical activity score was calculated for each amenity building by summing the number of elderly participants doing a certain activity multiplied by the METs value for that activity. For example, if there were three old people sitting (MET = 1.0), two talking (MET = 1.5), and one exercising (MET = 5.0), the physical activity score for the amenity building would be: 3 × 1 + 2 × 1.5 + 1 × 5.0 = 11. The calculation formula is as follows:(1)$$\sum_{n=x}^{y}=N_{n}*MET_{n},$$
‘n’ represents a type of activity, ‘*N*’ represents the number of elderly participants of the activity, and MET represents the physical energy consumption value of the activity.

## 3. Results

Through these observational, scoring, and questionnaire processes, a total of 728 activity statistics were recorded, and 387 questionnaires were collected, among which 270 were found to be valid. The 117 invalid questionnaires included 104 questionnaires from residents who were found to be under the age of 50, as well as 13 questionnaires which were filled-in incomplete.

### 3.1. Participant Characteristics

The data of behavioral observations mainly recorded content related to activities, while the questionnaire collected the study participants’ background information. As such, the recorded characteristics of the participants were mainly based on the questionnaire data. The profiles of the participants are shown in Table 3 below. Males accounted for 43.5% and females for 56.5% of the total questionnaire group, with their ages mainly ranging from 60 to 69 years old, accounting for nearly half of the participants. The occupations of the participants, prior to retirement, covered a wide range of industries, of which more were general staff and professionals, followed by general workers and business managers. When asked whether they would usually choose to go to an amenity building, such as a pavilion, as a place for leisure every time when going out for activities, 76.4% of older adults responded yes. Note also that amenity buildings are free to use in China.

### 3.2. Health Activity Characteristics

#### 3.2.1. Activity Frequency and Duration

There was a rich variety of activities that elderly groups participated in at the amenity buildings, mostly consisting of leisure and entertainment activities, including chatting, sitting and watching, playing chess and cards, singing, etc. Except sitting alone, all other observed activities have the characteristics of a social interaction.

Statistics on activity frequency (Figure 5a) show the elderly group’s relative activity preferences associated with amenity buildings. The most frequent activities included chatting (33%), playing table games (e.g., chess and cards) (20%), sitting and watching (18%), and music-related activities (12.2%). It was observed that those older adults who had much experience visiting the amenity buildings were able to easily sit down and find appropriate topics to discuss with each other, which also supports the idea that the elderly population’s social interactions are mainly based on verbal communication.

Playing table games and music-related activities were the two social activities with the largest number of participants. Music-related activities included singing songs, Chinese opera, and playing musical instruments. It is noteworthy that for both of these two types of activities, table games and music, often gather larger groups of people and reflect positive social interactions (Figure 6). Playing tables games requires intense mental activity, while music is usually an emotional and light physical activity. As such, the two most popular activities appear to be those that provide great forms of encouragement to older adults’ health, both physically and mentally.

According to the data regarding different activities duration and typical time periods (Figure 5b), four activities were noted as occupying all five main time periods of the day: chatting, physical activity, dancing, and playing instruments. Chatting was observed as being the activity with the highest degree of continuity and frequency. This is understandable as chatting is the most basic form of social communication for older adults, but the activity’s value should also be appreciated as it can stimulate an individual’s thinking, which is beneficial to both the emotional communications among older adults and to helping them maintain a good mental state [80]. It can be seen from the data that older adults chat throughout almost the entire day while the amenity building provides a special place for them to do so, surrounded by a beautiful landscape, and in turn, this space and landscape may be well-suited for elderly groups to adjust their body and mind. Additionally, it was found that playing table games required proper lighting, so these activities usually only took place from 10:00 to 18:00. In contrast, due to the spatial features provided by amenity buildings, singing, dancing, and playing instruments can be performed almost all day, except during the noon break.

#### 3.2.2. Activity Restoration Effects

According to the World Health Organization, health is a state of complete physical, mental, and social wellbeing, not merely the absence of disease or infirmity [81]. A main focus of the questionnaire was the restoration effects the elderly participants self-reported following their engagement in social activities. This included six dimensions: cheerfulness, stress relief, physical fitness, social interactions, reducing loneliness, and enhancing self-worth (Figure 7). As can be seen from the chart (Table 4), the two factors of physical fitness and cheerfulness had the highest scores, with average values of 3.98 and 3.92, respectively. This represents the basic feelings participants had following their activities, and also indicates the unique advantages of the amenity building and park environment. The next two highest average values were for stress relief and reducing loneliness, which were 3.83 and 3.85, respectively. This further indicates that social interactions can help greatly reduce the loneliness and stress associated with older adults. The average scores of social interactions and enhancing self-worth were 3.69 and 3.64, respectively. We believe that assessing these two items requires a certain degree of self-analysis due to them being relatively abstract. Therefore, we believe older adults felt less strongly about these two dimensions compared to the four others. Regardless the scores remained high, indicating that social activities have a positive effect and play a positive role in helping regulate the HRQoL of older adults.

In Table 4, we also include the questionnaire results for those respondents under the age of 50 for comparison. The two highest mean scores were for cheerfulness and stress relief, at 3.87 and 3.82, respectively. The mean score of physical fitness was 3.67, and reducing loneliness was at 3.33 points. The two items with the lowest mean scores were social interaction and enhancing self-worth, at 3.18 and 3.10, respectively. The variance of all six items is generally high, between 1.5 and 3.

### 3.3. Role of Media and Site Advantages of Amenity Buildings

Our study’s statistics show that older adults often go to the park alone, and this accounts for 12.8% of all visitors, while also highlighting their loneliness. The proportion of people traveling to the park in a group of two people was 36.3%, with most of them coming as couples. Those coming with a group of three or more accounted for 50.8% of the remaining total. This also matches a traditional saying in China that, “Three to five people always make groups”. For those involved in group activities with three or more people, the questionnaire asked further information regarding the way in which the participant first learned about and joined their current activities, and thereby this helps us better understand the source of social behavior.

According to the associated data (Figure 8), it was found that about 38.5% of such group activities were recommended to the participant by a friend. Another 39.6% of older adults encountered the activities by chance while visiting the parks and then joined later on. Only 8.8% of elderly participants had discovered or joined such a group via online communities and platforms, such as WeChat. This finding further indicates that the role of online platforms in helping form social connections among older adults population is very limited, while, in contrast, an amenity building with a beautiful landscape may become the basis of a strong social media.

When asked whether amenity buildings would increase the elderly participant’s willingness to go out on a rainy day, 60.2% of the participants said yes and 39.8% said no. This finding shows that the shelter and safety of amenity buildings provides for the possibility of older adults to go out on a rainy day.

### 3.4. Correlation Between Amenity Building Factors, Number of Activity Participants, and Physical Activity Score

Below, we have documented all the salient statistical information regarding our correlation analysis between the identified factors and number of activity participants and physical activity scores. Note that unless otherwise specified, the “number of activity participants” here is referring specifically to the number of older adults identified participating in the activity. That is, activity participants under 50 are not included in this data.

When reviewing the surrounding environment factor’s three sub-indexes (Table 5), several points were noted. First, the correlation coefficients of water features with the number of activity participants (0.372) and physical activity score (0.279) were both low. Similarly, the correlation coefficients of vegetation with number of activity participants and the physical activity score were 0.277 and 0.239, respectively, also relatively insignificant. However, in contrast, the CSE factor was positively correlated with both the number of activity participants and the physical activity score, with corresponding R-values of 0.519 (*p* < 0.01) and 0.446 (*p* < 0.05). This indicates that the CSE factor has a significant positive effect on the number of participants, with a high degree of influence. In addition, the CSE factor also has a significant positive influence on the physical activity score, but the degree of influence is only moderate.

Among the three sub-indexes of the Amenity Conditions factor, scale was positively correlated with the number of activity participants and physical activity score, with R-values of 0.635 and 0.524, respectively, both of which are significant at the 0.01 level. The scale of an amenity building indicates the maximum capacity of people the structure can hold, so this naturally also has a large impact on the activities of older adults.

The surrounding amenities factor was positively correlated with both the number of activity participants and physical activity score, and had a significant degree of influence (*p* < 0.05). The R-values of corresponding correlation coefficients are 0.414 and 0.439, representing a moderately high degree of influence. Meanwhile, the standard deviation of the surrounding amenities is 0.917 and the average score is 2.21, which is very low, indicating that there is basically no infrastructure in the amenity building. To a moderate extent, the surrounding amenities had positive effects on the number of activity participants and physical activity score.

The Pearson correlation value between lighting and the number of activity participants was 0.596, at a level of 0.01. The value between lighting and physical activity score was 0.458, which was significant at the 0.05 level. This shows lighting has a great influence on the recreational activities of older adults.

In the factor analysis of activity types, the mean score and standard deviation of activity types in amenity buildings were 4.07 and 1.464, respectively. Basically, this indicates a variety of activities will take place at all the amenity buildings. The correlation coefficients of activity type with number of activity participants and physical activity score were 0.775 (*p* < 0.01) and 0.660 (*p* < 0.01), respectively. The number of activity types performed by older adults has a significant positive correlation with the number of activity participants and physical activity scores of amenity buildings. In other words, when an amenity building is associated with larger numbers of participants or a higher activity score, there will tend to be a higher number of activity types based at the amenity building.

### 3.5. Relationships Between Factors of Amenity Building and Different Activity Types

The Environmental Index’s total score (EI) is a composite score of the Surrounding Environment and Amenity Conditions factors, thereby integrating all of the sub-indexes into one total score. Data analysis (Table 6) of this shows there is a statistically significant correlation between various activities and the EI score of amenity buildings (X2 = 615.8, df = 42, *p* < 0.001). The table shows the number and percentage of elderly activity participants, as well as adjusted standardized chi-square residuals for each activity and building. Adjusted standardized residuals of +2.0 or greater (marked in green) indicate a higher number than expected and standardized residuals of −2.0 or less (marked orange) indicate a lower number than expected.

Among the seven typical activities of elderly groups, sitting and watching, standing, and chatting are assumed as basic leisure and social activities, while table games, music-related activities, and dancing are all the activities with significant entertainment value and social interaction.

The activities of sitting and watching and standing were more likely to take place in amenity buildings with a high EI score. The proportion of 22 points was significantly higher than expected (the residual was greater than +2.0), indicating that older adults were more likely to sit and stand in amenity buildings with higher environmental quality.

The majority of the chatting activities occurred in amenity buildings with EI scores between 12 and 17 points, while the proportion of 18 and 22 points was significantly smaller than expected (the residual is less than −2.0). This indicates that there is basically no environmentally related demand for older adults to engage in chatting. Chatting is a leisure activity and more focused on communication. The EI scores of all activities regarding table games in the amenity buildings are nearly all between 17 and 19 points, indicating that older adults are pickier about the environmental quality of the space when playing such games, and when in a low quality environment, they will hardly carry out any such activities.

The proportion of music-related activities was mostly in amenity buildings with EI scores of 22, which was significantly higher than expected. This indicates that older adults prefer to engage in music-related activities in high-quality environments. For the high-intensity activity of dancing, the environmental index of all samples is between 16 and 17, indicating that older adults have limited or no requirements of the environment when dancing. Combined with a consideration of China’s national conditions, this finding also shows that the significance of dancing is primary for exercise and social interaction [82], which may make dancing itself a kind of cultural landscape, with low requirements for the external environment.

Overall, the study’s analysis shows that the elderly visitors of Nanjing’s public parks tend to choose amenity buildings with high environmental quality when carrying out social activities such as sitting and watching, playing table games, and dancing, while they have the highest requirements for environmental quality when engaging in music-related activities.

## 4. Discussion

### 4.1. Health Benefit of Amenity Buildings for Older Adults

According to the results of the self-assessment questionnaire, social activities can play a positive role in six dimensions of their physical, psychological, social, and self-cognition, including cheerfulness, stress relief, physical fitness, social interaction, reducing loneliness, and enhancing self-worth. As a primary site of such activities, amenity buildings not only provide a safe place for older adults to shelter from rain, but also provide them a venue that enables and enhances these activities with a high landscape index.

The effects of social activities on these six dimensions for the elderly population were ordered from high to low according to their mean scores among the demographic: physical fitness, cheerfulness, reducing loneliness, stress relief, social interactions, and enhancing self-worth. By contrast, the scores and rankings of the same six dimensions of people under 50 were significantly differently from the elderly population: cheerfulness, stress relief, physical fitness, reducing loneliness, social interactions, and enhancing self-worth.

These differences begin to show how elderly and non-elderly groups vary in their relation to public spaces and opportunities to use the spaces to socialize or exercise. In terms of these comparative rankings, physical and mental health ranked first for the elderly group, but for those under 50 years old who have not retired, cheerfulness is less of a priority. Due to physiological reasons, the physical strength of elderly citizens generally declines with time, and so there is a natural difference between the groups. Walking becomes basic aerobic exercise for older adults as it allows them to strengthen their bodies [83]. Long work hours were revealed to be negatively related to overall levels of physical activity, which was a more influential physical determinant for non-elderly people than older adults [84]. These different groups subsequently may visit the park for different purposes. For older adults, the most basic purpose and appeal of going to the park is to exercise, while for other groups, going to the park is more to relax and watch the scenery, and the park is not primarily a place to exercise for them.

There was no significant difference between the elderly and non-elderly groups in the two dimensions’ scores of cheerfulness and stress relief, both which stood at close to 4 points for both groups. This indicates that engaging in public park activities has a significant psychological restoration effect on all age groups. Another interesting finding was that cheerfulness and stress relief were the two dimensions with the highest scores reported by people under 50, but ranked second and fourth, respectively, among the elderly group, suggesting that the pressure of the elderly group is lower than the non-elderly group to some extent.

In terms of restoration effects, the average self-reported score of the elderly group was significantly higher than that of the non-elderly groups for the three dimensions of social interaction, reducing loneliness, and enhancing self-worth, with an approximate 0.5 score difference across all three. This indicates that in terms of socializing, amenity buildings provide the greatest support and have the most value for older adults as they significantly help promote social interactions and social connections among older adults more than any other groups. As such, older adults in China prefer to stay in amenity buildings when visiting parks.

These findings suggest that amenity buildings serve a dual role for the elderly population, both as an exercise space for primarily low-intensity activities and an important social space for seeking social connections. Early studies noted that public parks function as public spaces for older adults to socially interact, and help sustain their social lives [33,85,86]. A previous study also suggested that older adults regard such open spaces as gathering spaces and as a sign of social interaction [32]. It also has been reported that many physical environment determinants in open spaces have an impact on the physical activities of older adults [87]. At the neighborhood level, a negative association was reported for negative street characteristics (such as lack of sidewalks and streetlights). At the macrolevel, environment score and environmental barriers were all positively associated with overall physical activities. The research scope of this study was not a single park or neighborhood community, but the subjects were primarily older people from the residential areas of Nanjing’s traditional urban district, which are near parks.

Due to the limited public and green spaces in these communities, the amenity buildings in parks constitute places for social interaction among older adults. They have even become known as a “natural neighborhood network” [31] with Chinese characteristics.

Amenity buildings provide an important and more stable place for older adults to socialize, especially for leisure-based social activities, such as sitting and chatting, and gathering-based social activities, such as playing table games and doing music-related activities.

From the perspective of activities, table game activities are highly frequent among older adults in amenity buildings, with large audiences often gathering to spectate and comment. There are several reasons for this. Unlike other park spaces, amenity buildings provide a stable structure for playing such games. The structure’s seats not only can be used as seats, but sometimes serve as tables. Relatedly, older adults can also use the structure to avoid disruptive weather, such as wind, rain, and intense sunlight. Activities like chess and cards are simple, low-intensity mental activities, which are favored by older adults. The number of spectators often exceeds the number of players, which also shows this culture’s high popularity in China. Even in winter, older adults can be seen holding thermoses to keep warm and gathering in amenity buildings for recreation. Based on this phenomenon, it seems higher requirements must be put forward for the construction of such amenity buildings. The seating capacity of existing amenity buildings can only meet the basic needs of chatting and rest, while the structure’s support for social activities such as playing chess and cards is clearly wanting.

Music-related activities are another elderly activity with regional characteristics. Although most older adults are not professional musicians, their performance is more for leisure and entertainment. Older adults come to sing songs and rehearse operas, and in turn, this can help regulate their emotions and behaviors and reduce psychological stress. In addition to mental health, playing musical instruments also has varying degrees of impact on physical health. Playing highly demanding music performances will have a greater impact on human cardiovascular health, compared to a moderately demanding performance [88]. It has been suggested that continuing to play music during the aging process can be physically sustaining and cognitively beneficial. Meanwhile, older adults should pay attention to the time and intensity of playing instruments, as well as their physical conditions, to avoid overload [89]. Relatively speaking, it is also worth mentioning that amenity buildings usually have various types of sloping roofs, and the space under these sloping roofs can improve the sound quality and reverberations to a certain degree [90,91]. Clearly, such space better supports these activities than the park’s other open spaces.

Unlike waterfront restaurants, cafes, or other buildings, the free and open design of amenity buildings in city parks does not usually have clear functional orientations. Instead, activities are spontaneously organized and formed through people’s participation. However, such spontaneous activities can only occur when the external conditions are appropriate and the site is attractive [38]. As such, the social utility of the amenity buildings greatly depends on its external and surrounding environment. A previous study suggested that providing appropriate outdoor spaces for informal socializing may attract older adults into more active lifestyles [92]. Supporting this notion, all the above findings indicate that amenity buildings in city parks contribute to increasing outdoor social activities among older adults. This is of great value in a rapidly aging society due to how such spaces can positively help older adults in several psychological, physiological, and social aspects.

### 4.2. Amenity Buildings’ Positive Influence Factors and Their Enhancement

The exploration of the amenity building’s typical physical factors can help us further understand how to build a high-quality amenity building and its associated public park environment. This study’s correlation analysis showed that six such physical factors (water features, vegetation, CSE, surrounding amenities, scale, and lighting) all relate to the activities of older adults. These factors were also found to all positively correlate with the number of elderly activity participants and physical activity score. The number of participants generally reflects the comprehensive attraction of the amenity building to older adults, and the physical activity score partially indicates the physical activity levels and physical health benefits of the elderly population of that site. The physical activity score also explains the preference of physical activities among older adults and the preferences for amenity buildings. The Environment Index (EI) score was designed to capture the overall impact of the environment on a given amenity building site. The subsequent Chi-square statistical analysis of these data showed that the social activities of older adults are significantly correlated with a given site’s overall EI score, but the score has different levels of influence towards different types of social activities.

In order from high to lowest correlation, the factors that correlated the closest with numbers of activity participants and physical activity score were ranked as follows: activity type, scale, lighting, CSE, surrounding amenities, water features, and vegetation.

Activity type was the most correlated factor, and this reflects the degree of support and development trends an amenity building has to host diverse activities.

The scale of an amenity building was the second most impactful factor (0.635), and this reflects the space capacity limitations in terms of number of simultaneous visitors. This, in turn, affects the attractiveness of the amenity building to a certain extent.

The lighting arrangements of amenity buildings also had a high impact on elderly activities. Offsetting corresponding natural declines in visual abilities among older adults, proper lighting provides conditions for nighttime activities and ensures visitors’ safety. The results are consistent with earlier studies, showing the importance of good maintenance and management of public parks, including cleanliness and adequate lighting to ensure safe walking and sitting [8,93].

The fourth highest correlated factor was the CSE (0.519). The higher this factor, the higher the quality an amenity building’s landscape and surrounding environmental elements must be, and thereby the more attractive it is to visit and engage in activities for potential elderly visitors. This finding, supporting the importance of environment factors, confirms what previous studies have shown, which is the critical role natural elements play in making leisure-based environments be considered attractive [53,94]. It shows that comprehensive environmental factors are not only of aesthetic significance, but also of social significance. In other words, attractiveness is important, as it implies visitors’ good opportunities for social interaction with friends or strangers, and this is also an important aspect in the life of older adults.

Surrounding amenities was the fifth most relevant factor, but it still had relatively high R-values of 0.414 and 0.439. The average score of this factor among sites surveyed was low, indicating that generally, these sites only have seats, and amenities such as tables, drinking fountains, and toilets remain quite insufficient. In particular, as people grow older, their physical function gradually declines and walking speeds become almost half of other non-elderly adults, while simultaneously, their frequency for needing to go to the bathroom increases. The availability of a toilet within 250 m of an amenity building, therefore, becomes an important infrastructural indicator under the elderly-oriented criteria. Consistent with previous research, more social activities occur in places with good amenities and accessible bathrooms because the burden of an older person leaving their home is reduced if a restroom is readily available at their destination [92]. In addition, when faced with insufficient infrastructure support, elderly visitors may also spontaneously bring their own items, such as folding tables and chairs, music shelves, and even lamps, to fulfill to their own activity needs. This behavior highlights the mismatch that exists between the reality of existing amenity buildings and their visitors’ demands. As such, it is suggested that future amenity buildings should be designed and set up according to the needs and activity preferences of older adults.

The two factors with the lowest correlations were water features and vegetation, of which vegetation had the lowest value of 0.277, implying no significant correlation. One possible reason is that in the landscape environment, vegetation elements are already ubiquitous, so including more vegetation into the amenity building’s environment may not stand out, nor have any tangible impact on the social activities of older adults. However, an interesting finding is that although the individual correlation coefficients of water and vegetation were low, the correlation of the CSE factor (water + plants + special landscape) was significant. This may indicate that older adults tend to prefer to engage activities in amenity buildings with a high level of integrated environmental quality, rather than any singular environmental element.

In summary, these findings suggest that to increase the social utility of amenity buildings for elderly populations, by having such spaces encourage more social interactions and outdoor exercise, the quality of the amenity building’s surrounding landscape environment should be improved, the building areas should be appropriately expanded, and the surrounding amenities and lighting arrangements should also be increased.

Through comparing the total score of Environment Index and the activity types of the study’s ten amenity building sites, the demand tendencies of their visitors on the environment as related to different activities can be explored. It was found that 43.8% of the elderly visitors were more likely to sit and enjoy the scenery in an amenity building of higher environment quality. However, when chatting, the number of older adults is concentrated in the middle range of the score, which does not have high requirements for environment. The possible reason is that chatting is not usually a purposeful activity for older adults and can happen almost anytime and anywhere, with little dependency on the features of the environment. In comparison, when engaging in table game activities, such as chess and cards, or music-related activities, older adults tend to choose amenity buildings with high scores of environment factors. Such table game activities have higher requirements on the amenities provided by the environment because the activity requires staying in one place for a long time, so amenities such as tables, seats, and toilets become more necessary. Both exercise and dance belong to social activities with medium and high intensity. Among them, the form of dancing found in China’s public spaces is a collective social exercise activity with Chinese characteristics, which usually requires a large scale space but limited amenities, so its requirements on the Environment Index are in a medium position.

Based on the above analysis results, when designing amenity buildings, the activity preferences and physiological conditions of older adults should be considered as a priority. With this understanding, such spaces can improve their functional comfort and location layout, and offer more support to specific social activities.

## 5. Conclusions

Urban green spaces have become the main outdoor activity site for China’s growing elderly populations. In Nanjing, this is especially true in the city’s traditional urban districts along the ancient city wall. In the current urban green spaces of Nanjing, amenity buildings such as pavilions and shelters, which provide a free space, play an equivalent social role as traditional neighbor communities. They provide basic space and guarantees for the stable social activities of older adults and have become an important “third place” in citizens’ daily lives. Among older adults, the occurrence, participation, and continuity of social interactions in beautiful, natural environments play an important and positive role in maintaining their physical and mental health, as well as their sense of social connections for older adults, and has the meaning of a “natural neighborhood network” [31].

As mutual contact among older adults is the basis for the formation of social ties, social activities that the elderly visitors engage frequently at a city park’s amenity buildings play a fundamental role. Select activities include chatting, playing chess, sitting and watching, music-related activities, dancing, etc. Participation by older adults in such activities in the parks’ landscape environment has shown to significantly improve all aspects of their health and wellbeing, especially in the sense of social connections and reducing loneliness as compared with other non-elderly demographics. Combining six physical indicators of the amenity buildings, correlation analysis and Chi-square statistics were used to analyze the correlation between each factor and the number of elderly participants in these social activities, as well as physical activity score. This analysis revealed the potential relationships between the total Environmental Index and different types of activities.

The results showed that activity types, scale, lighting, CSE, surrounding amenities, water features, and vegetation were positively correlated with attracting older adults to engage in social activities. Among them, scale, lighting, CSE, and surrounding amenities were physical factors with significant influence on their engagement. As for activity types, the relationship between different activity types and the total Environmental Index varies considerably. For example, low-intensity social activities, such as chatting, and table game activities have moderate EI demands, while sitting and watching and music-related activities have a high EI demand.

The results of the correlation analysis show that the amenity building’s scale, lighting arrangements, CSE, surrounding amenities, water features, and vegetation all have a high influence. In particular, the correlation value of the CSE factor indicates that older adults are more sensitive to the overall quality of the environment rather than any particular environmental element. Subsequently, urban parks near communities can be considered a positive supplement to the neighborhood community environment to enhance the functional amenity building. To improve their societal utility for the elderly populations, the location and layout of amenity buildings should be designed with consideration towards the needs of older adults. As such, the design trend of amenity buildings can achieve a dual nature of “leisure + society”, and under an accelerating aging trend, amenity buildings can become an even more important infrastructure within urban parks and their surrounding cities.

This study has provided empirical data on the social interactions among older adults and provides theoretical support and reference for the future development of amenity buildings which are oriented towards the needs of older adults. As such, we believe these results may be of significant value towards improving open urban spaces which have primarily elderly demographics, and more generally, the living conditions of traditional urban spaces with predominantly elderly residents.

## Figures and Tables

**Figure 1 ijerph-17-07497-f001:**
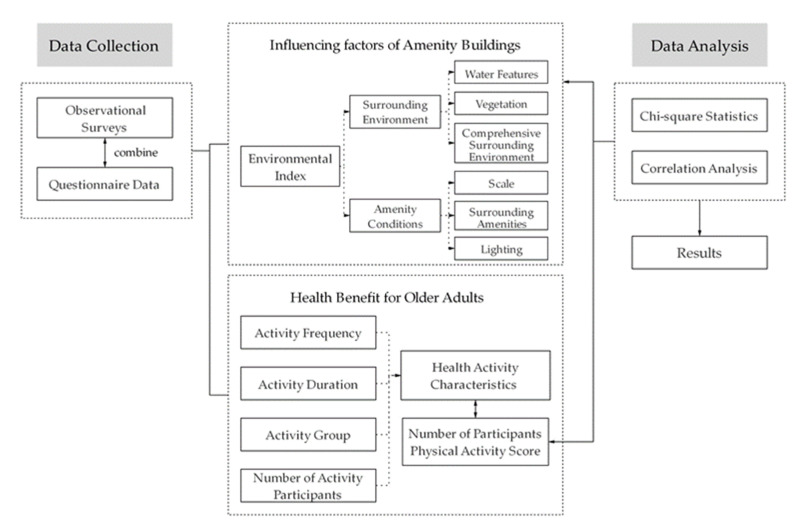
Research plan overview.

**Figure 2 ijerph-17-07497-f002:**
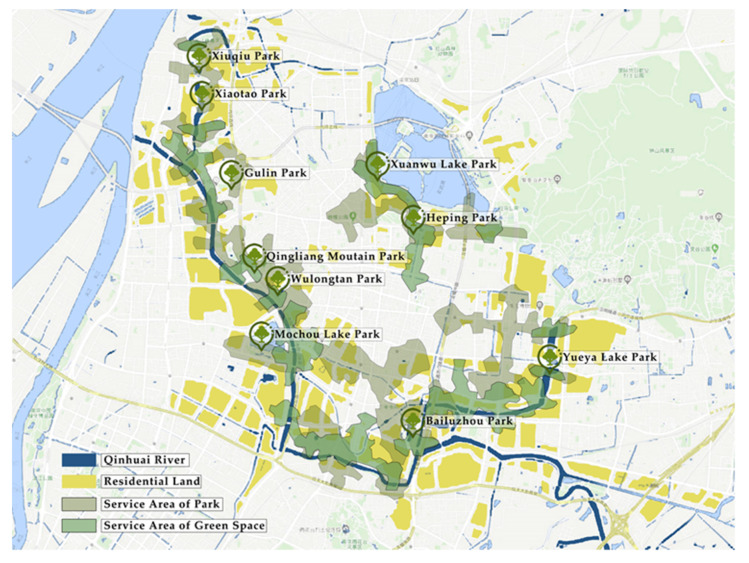
Geographic distribution of parks surveyed.

**Figure 3 ijerph-17-07497-f003:**
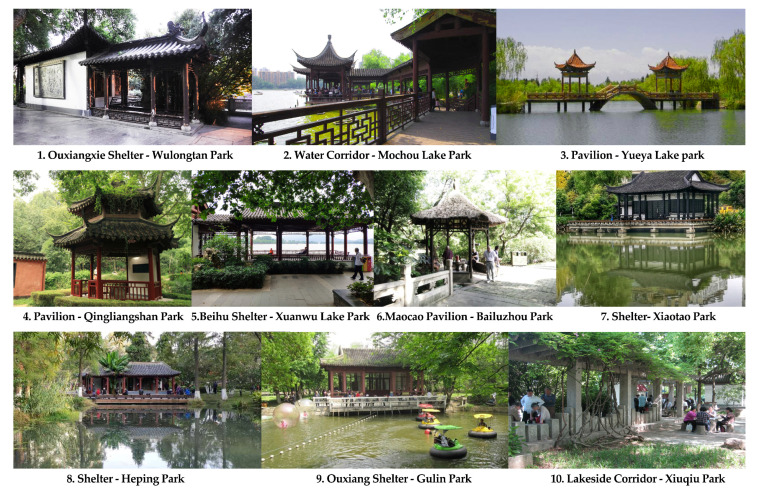
Photos of the 10 amenity buildings included in the study.

**Figure 4 ijerph-17-07497-f004:**
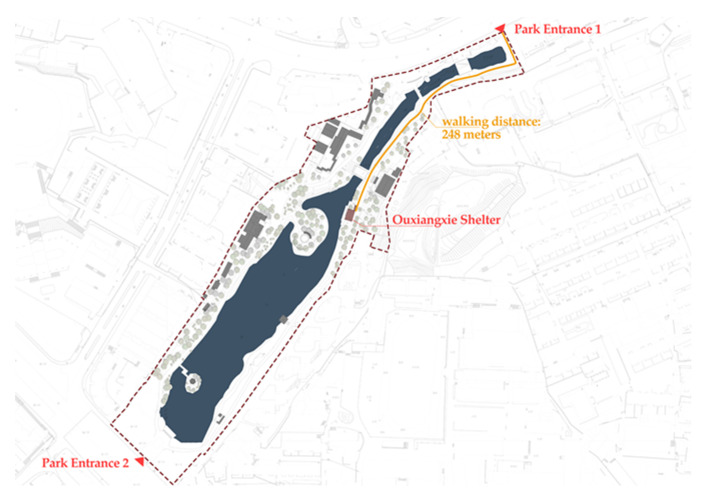
Example: Location of Ouxiangxie Shelter in Wulongtan Park.

**Figure 5 ijerph-17-07497-f005:**
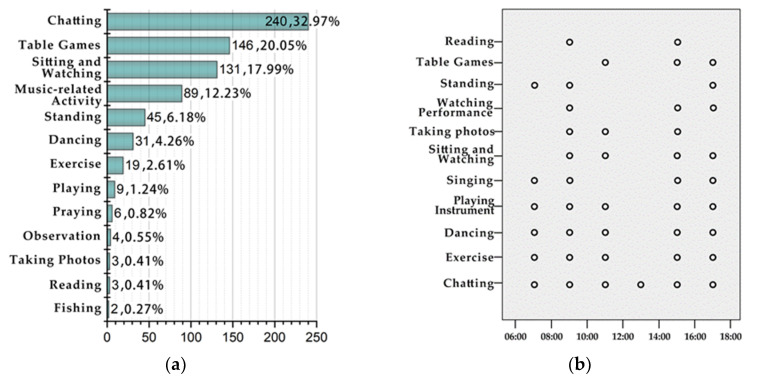
(**a**) Frequency of activities; (**b**) Duration of activities.

**Figure 6 ijerph-17-07497-f006:**
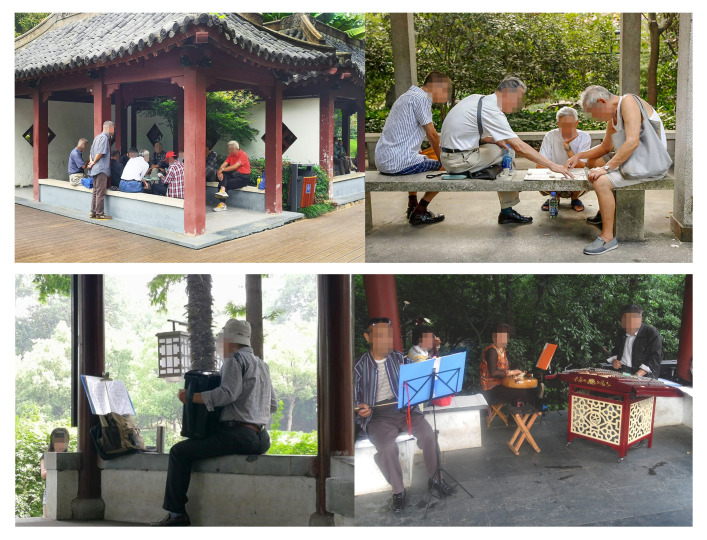
Older adults engaging various activities in amenity buildings.

**Figure 7 ijerph-17-07497-f007:**
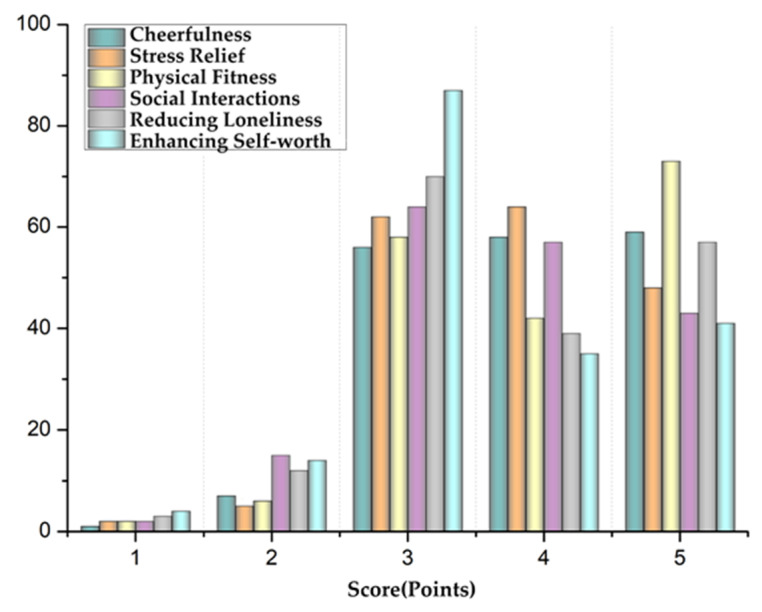
Self-reported post-activity restoration effects.

**Figure 8 ijerph-17-07497-f008:**
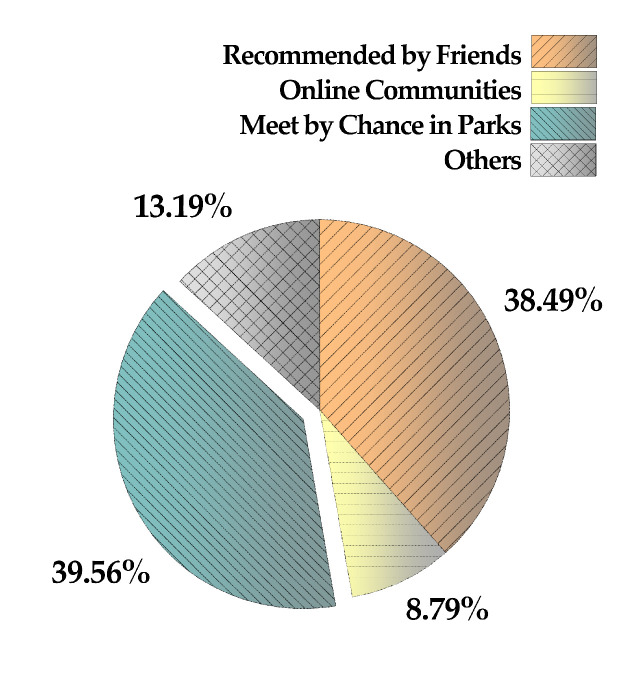
Channels used for joining elderly group activities.

**Table 1 ijerph-17-07497-t001:** Measured factors and weight [59].

Physical Factors			Scores		
**Environ-mental Index**	**Surrounding Environment**	Water Features	Water Enclosures	no water=0; <10% = 1; 10–20% = 2; 20–30% = 3; 30–40% = 4; 40–50% = 5; 50–60% = 6; 60–70% = 7; 70–80% = 8; 80–90% = 9; 90–100% = 10	
		Vegetation	Vegetation Density	no vegetation = 0; <10% = 1; 10–20%= 2; 20–30% = 3; 30–40% = 4; 40–50% = 5; 50–60% = 6; 60–70% = 7; 70–80% = 8; 80–90% = 9; 90–100% = 10	
		Comprehensive Surrounding Environment (CSE)	Water Features	The average score of water and vegetation (Bonus points: one point for each special landscape feature, such as birdsongs and fountains, with a high score cap of 10)	
			Vegetation		
	**Amenity Conditions**	Scale	<10 m^2^ = 1; 10–20 m^2^ = 2; 20–30 m^2^ = 3; 30–40 m^2^ = 4; 40–50 m^2^ = 5; 50–60 m^2^ = 6; 60–70 m^2^ = 7; 70–80 m^2^ = 8; 80–90 m^2^ = 9; >90 m^2^ = 10		
		Surrounding Amenities	Seat	None = 0, one = 1, two = 2, three or more = 3	Combination of all scores
			Table	Withoutt = 0, with = 1	
			Rubbish Bin	Without = 0, with = 1	
			Vending Machine	Without = 0, with = 1	
			Drinking Fountain	Without = 0, with = 1	
			Storage Box	Without = 0, with = 1	
			Bulletin Board	Without = 0, with = 1	
			Toilets (within 250 m)	Without = 0, with = 1	
		Lighting	Lighting Inside Corridor	None = 0, weak = 1, strong = 2	Combination of all scores
			Garden Lighting	None = 0, weak = 1, strong = 2	
			Lawn Lighting	None = 0, weak = 1, strong = 2	
			Building Contour Lighting	None = 0, weak = 1, strong = 2	
			Nearby Street Lighting	None = 0, weak = 1, strong = 2	
**Activity Factors**			**Calculation basis**		
**Activity** **Index**	Types of Activity		Number of Activity Types		
	Number of Activity Participants		Number of Elderly Participants		
	Physical Activity Score		A physical activity score was calculated for each amenity building by summing the number of elderly participants doing a certain activity multiplied by the metabolic equivalents of energy (METs) value for that activity.		

**Table 2 ijerph-17-07497-t002:** METs value of select amenity building activities.

Intensity	METs	Activities of Older Adults	Other Activities
Low	<3	Sitting 1.0, Standing 1.2, Taking photos 1.3, Observing nature 1.3, Reading 1.3~1.8, Chatting 1.5, Playing chess and cards 1.5, Fishing 2.0, Playing Instruments 2.5	Practicing calligraphy 2.0
Medium	3~6	Praying 3.0, Singing 3.0, Singing opera 3.0, Childcare 3.5, Exercising 5.0	Walking 3.0, Dog walking 3.5, Playing badminton 4.5, Bicycling 6.0
High	>6	Dancing 7.0	Jogging 7.0, Practicing martial arts 10.0

**Table 3 ijerph-17-07497-t003:** Basic demographic features of questionnaire respondents.

Basic Features		Proportion	Basic Features		Proportion
Gender	Male	43.5%	Career	Civil Servant	7.6%
	Female	56.5%		Business Manager	14.3%
Age	50–59	32.5%		General Staff	22.8%
	60–69	43.5%		General Workers	14.8%
	70–79	19.4%		Self-employed	2.5%
	80+	4.6%		Freelancer	5.5%
Whether amenity buildings will usually be chosen as a place to go when out for activities	Yes	76.4%		Business Services Workers	3.0%
				Farmers	1.7%
	No	23.6%		Other	3.8%
				Professionals (e.g., Doctor/teacher)	24.0%

**Table 4 ijerph-17-07497-t004:** Activity restoration effects.

Restoration Items	Over 50 Years Old				Under The Age of 50			
	Mean	SD	Variance	N	Mean	SD	Variance	N
Cheerfulness	3.92	0.92	0.94	270	3.87	1.24	1.56	104
Stress Relief	3.83	0.89	0.79	270	3.82	1.29	1.68	104
Physical Fitness	3.98	0.98	0.96	270	3.67	134	1.79	104
Social Interaction	3.69	0.96	0.93	270	3.18	1.54	2.39	104
Reducing Loneliness	3.85	1.03	1.06	270	3.33	1.53	2.34	104
Enhancing Self-worth	3.64	0.99	0.98	270	3.10	1.69	2.87	104

**Table 5 ijerph-17-07497-t005:** Correlation between factors and number of activity participants and physical activity score.

Factors R: Pearson Correlation, S: Significance (Double Tails)				Number of Activity Participants	Physical Activity Score	Descriptive Statistics	
						Average	SD
**Environmental Index**	**Surrounding Environment**	Water Features	R	0.372	0.279	2.54	1.71
			S	0.052	0.150		
		Vegetation	R	0.277	0.239	5.57	2.04
			S	0.154	0.221		
		CSE	R	0.519 **	0.446 *	8.11	2.17
			S	0.005	0.017		
	**Amenity Conditions**	Scale	R	0.635 **	0.524 **	2.89	1.31
			S	0.000	0.004		
		Surrounding Amenities	R	0.414 *	0.439 *	2.21	0.92
			S	0.029	0.022		
		Lighting	R	0.596 **	0.458 *	1.43	1.37
			S	0.001	0.014		
**Activity Index**	Activity Types		R	0.775 **	0.660 **	4.07	1.46
			S	0.000	0.000		

** The correlation was significant at 0.01 level (double tails). * Correlation was significant at 0.05 level (double tails).

**Table 6 ijerph-17-07497-t006:** Cross-tabulation of scored influencing factors (environmental index) by activity type.

Activity Type		Environmental Index Score ^1^ of Ten Amenity Buildings ^2^										Total
		No. 3	No. 4	No. 7	No. 1	No. 9	No. 5	No. 8	No. 10	No. 6	No. 2	
		9	10	10	12	16	17	18	19	22	22	
Low intensity	Sitting and Watching	5	0	7	14	20	21	3	23	10	28	131
		3.8%	0.0%	5.3%	10.7%	15.3%	16.0%	2.3%	17.6%	7.6%	21.4%	18.7%
		1.3	1.3		0.6	1.3	−2.9	−3.4	1.8	2.4		
	Standing	4	0	0	3	7	12	0	4	9	6	45
		8.9%	0.0%	0.0%	6.7%	15.6%	26.7%	0.0%	8.9%	20%	13.3%	6.4%
		3.1	−1.2		−0.5	0.8	−0.4	−2.4	−0.6	2.1		
	Chatting	5	0	9	36	41	107	2	19	10	11	240
		2.1%	0.0%	3.8%	15.0%	17.1%	44.6%	0.8%	7.9%	4.2%	4.6%	34.2%
		−0.1	0.4		3.1	2.6	4.2	−5.2	−1.8	−3.8		
	Table Games	0	0	0	0	0	22	85	25	0	14	146
		0.0%	0.0%	0.0%	0.0%	0.0%	15.1%	58.2%	17.1%	0.0%	9.6%	20.8%
		−1.8	−2.2		−3.6	−4.1	−3.2	15.3	1.8	−2.7		
Medium intensity	Music-related Activities	0	3	0	10	0	25	0	3	12	36	89
		0.0%	3.4%	0.0%	11.2%	0.0%	28.1%	0.0%	3.4%	13.5%	40.4%	12.7%
		−1.4	0.0		0.7	−3.2	−0.3	−3.4	−2.3	7.3		
	Exercise	1	4	0	0	0	2	0	10	0	2	19
		5.3%	21.1%	0.0%	0.0%	0.0%	10.5%	0.0%	52.6%	0.0%	10.5%	2.7%
		0.9	4.3		−1.3	−1.5	−1.5	−1.6	5.1	−0.9		
High intensity	Dancing	0	0	0	0	12	19	0	0	0	0	31
		0.0%	0.0%	0.0%	0.0%	38.7%	61.3%	0.0%	0.0%	0.0%	0.0%	4.4%
		−0.8	−1.0		−1.7	4.5	3.2	−2.0	−1.9	−2.5		
Total		15	7	16	63	80	208	90	84	41	97	701
		2.1%	1.0%	2.3%	9.0%	11.4%	29.7%	12.8%	12.0%	5.8%	13.8%	100.0%

^1^ Overall association is significant (X^2^ = 615.8, df = 42, *p* < 0.001). ^2^ The number of each amenity building matches Figure 3. Adjusted standardized residuals of +2.0 or greater (marked in green) indicate a higher number than expected and standardized residuals of −2.0 or less (marked in orange) indicate a lower number than expected.

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
