# Peer review of "The Social Utility and Health Benefits for Older Adults of Amenity Buildings in China’s Urban Parks: A Nanjing Case Study"

_ijerph, 2020, doi:10.3390/ijerph17207497_

Round 1

Reviewer 1 Report

Dear editor and authors, thank you very much for the opportunity to review this manuscript. The paper is original, well written, the statistical analyses is appropriate, however I have some minor corrections to suggest:

1 – Although still used, the term elderly seems to be outdated, please prefer to use “older adults”, “older men”, “older women”, “older persons” instead of elderly (refer to United Nations recommendations).

2 – I would strongly advise the author to modify Table 1, in order to present the data in a more clear way (e.g. reducing letter size).

3 - Revise the utilization of the term Health-Related Quality of Life along the manuscript. Please I advice the read of this paper: Karimi, M., & Brazier, J. (2016). Health, health-related quality of life, and quality of life: what is the difference?.Pharmacoeconomics, 34(7), 645-649.

4 - In the “Discussion” section, I would have wished to see more information on physical determinants of physical activity and its relationship with the results of this study. I recommend the next articles to improve lines (519-528; 543-552 and 569-574):

  • Carlin, A., Perchoux, C., Puggina, A., Aleksovska, K., Buck, C., Burns, C., ... & Coppinger, T. (2017). A life course examination of the physical environmental determinants of physical activity behaviour: a “Determinants of Diet and Physical Activity”(DEDIPAC) umbrella systematic literature review. PLoS One12(8), e0182083.
  • Puggina, A., Aleksovska, K., Buck, C., Burns, C., Cardon, G., Carlin, A., ... & Cortis, C. (2018). Policy determinants of physical activity across the life course: a ‘DEDIPAC’umbrella systematic literature review. The European Journal of Public Health, 28(1), 105-118.
  • Blasco-Lafarga, C., García-Soriano, C., & Monteagudo, P. Autonomic Modulation Improves in Response to Harder Performances While Playing Wind Instruments. Archives of Neuroscience, 7(2).

5 – Please revise this sentence (line 461-462): “Activity type has a significant positive influence on the number of activity participants and physical activity score at amenity buildings, and the correlation with is very high”.

I have no comments about the methodology of this study. The paper is well presented and I recommend it for publication after the previous suggestions.

Author Response

We gratefully thank you for spending your time making constructive remarks and useful suggestions, which has significantly raised the quality of the manuscript and has enable us to improve our research. We have studied comments carefully and have made correction which we hope meet with approval. Revised portion in the paper are highlighted using the "Track Changes" function in Microsoft Word. The main corrections in the paper and the responds to your comments are as flowing:

1.Comment: Although still used, the term elderly seems to be outdated, please prefer to use “older adults”, “older men”, “older women”, “older persons” instead of elderly (refer to United Nations recommendations).

1.Reply: We have made correction according to your comment and changed “elderly people” to “older adults/older persons” in the full text.

2.Comment: I would strongly advise the author to modify Table 1, in order to present the data in a more clear way (e.g. reducing letter size).

2.Reply: Thank you for your nice suggestion. We have made changes to Table 1 (line 248-249), including reducing the size of fonts, in order to present the data more clearly.

3.Comment: Revise the utilization of the term Health-Related Quality of Life along the manuscript. Please I advice the read of this paper: Karimi, M., & Brazier, J. (2016). Health, health-related quality of life, and quality of life: what is the difference?.Pharmacoeconomics, 34(7), 645-649.

3.Reply: It is really true as you suggested that the term Health-Related Quality of Life should be considered after reading the paper. So we have changed the words and sentences when describing the fourth part of questionnaire (line 221-223) and changed word “mental health” to “HRQoL” in result part when describing the results of self-reported restoration effects of older adults (line 396).

“And in the fourth section, a five-point Likert scale was used for their self-perceived health status after completing activities, as a measure of health-related quality of life (HRQoL) [58].” (line 221-223)

4.Comment: In the “Discussion” section, I would have wished to see more information on physical determinants of physical activity and its relationship with the results of this study. I recommend the next articles to improve lines (519-528; 543-552 and 569-574):

4.Reply: We gratefully appreciate for your valuable suggestions. And we have added some contents about physical determinants of physical activity according to the articles you mentioned about.

“Long work hours were revealed to be negatively related to overall levels of physical activities, which was a more influential physical determinant for non-elderly people than older adults [84].” (line 532-534)

“It also has been reported that many physical environment determinants in open spaces have an impact on physical activities of older adults [87]. At the neighborhood level, a negative association was reported for negative street characteristics (such as lack of sidewalks and streetlights). At the macro level, environment score and environmental barriers were all positively associated with overall physical activities.” (line 556-561)

“Music-related activities are another elderly activity with regional characteristics. Although most older adults are not professional musicians, their performance is more for leisure and entertainment. Older adults come to sing songs and rehearse operas, and in turn this can help regulate their emotions and behaviors and reduce psychological stress. In addition to mental health, playing musical instruments also has varying degrees of impact on physical health. playing highly demanding music performances will have a greater impact on human cardiovascular health, compared to mild performance [88]. It has been suggested that continuing to play music during the aging process can be physically sustaining and cognitively beneficial. Meanwhile, older adults should pay attention to the time and intensity of playing instruments, as well as their physical conditions to avoid overload [89].” (line 583-592)

5.Comment: Please revise this sentence (line 461-462): “Activity type has a significant positive influence on the number of activity participants and physical activity score at amenity buildings, and the correlation with is very high”.

5.Reply: We have re-written this part according to your suggestion. "The number of activity types performed by older adults has a significant positive correlation with the number of activity participants and physical activity scores of amenity buildings. Another words, when an amenity building is associated with larger numbers of participants or a higher activity score, there will tend to be a higher number of activity types based at the amenity building."(line 465-469)

Special thanks to you for your good comments.

Reviewer 2 Report

The authors presented an interesting topic that has an important implications in elderly population.

They carefully addressed an issue of the social utility and health benefits for older adults of amenity buildings in China.

Due to the rapidly aging population, the introduction of high-quality landscapes, adequate lightning, water features is of high importance.

It is very significant to encourage elderly people to take part in social activities, as it has a beneficial impact on mental health.

The work is well designed and very well described. Although, minor spelling English language corrections would be appreciated.

Additionally, I would advise authors to modify the Table 1 in order to make it more clear- reduce the font. After those minor corrections, I would accept this paper to be published.

I hope this response is satisfactory and will help the authors improve the manuscript.

Author Response

We gratefully thank you for spending your time reading our manuscript and making useful suggestions, which has enabled us to improve the manuscript. We have studied comments carefully and have made correction which we hope meet with approval. Revised portion in the paper are highlighted using the "Track Changes" function in Microsoft Word. The main corrections in the paper and the responds to your comments are as flowing:

1.Comment:  Although, minor spelling English language corrections would be appreciated.

1.Reply: We regret there were some problems with the English. The paper has been carefully revised by a native English speaker to improve the grammar and readability.

2.Comment: Additionally, I would advise authors to modify the Table 1 in order to make it more clear- reduce the font. 

2.Reply: Thank you for your nice suggestion. We have made changes to Table 1 (line 248-249), including reducing the size of fonts, in order to present the data more clearly.

Reviewer 3 Report

This study explores the influencing factors of such amenities and their behavioral affordances and based on the survey results, provides a theoretical basis for using these spaces towards the advancement of the elderly population. These days, especially, it is an extremely important study. I congratulate the authors for their creativity.

Author Response

We gratefully thank you for spending your precious time reading our manuscript and thank you very much for your recommendation.